# Decentralized Training of Transformer Models in Heterogeneous Network

## Abstract

Training large transformer-based models like GPT-4 and Llama3 is prohibitively expensive, often requiring vast resources, such as tens of thousands of GPUs running simultaneously for months. Traditionally, these models are trained in specialized clusters with high-speed, uniform interconnections and computational capabilities, enabling efficient data and pipeline parallelism. However, these clusters are costly, while more affordable GPUs are widely distributed across the globe. Existing approaches, such as Swarm and Dapple, primarily focus on distributed learning across data centers. In this paper, we introduce a novel framework designed to handle heterogeneous devices and unstable communication environments. Our framework employs a hybrid approach, combining parameter server architectures, pipeline parallelism, and task pool strategies to effectively manage device disconnections. Through comprehensive time-cost analysis and graph clustering techniques, we derive a near-optimal resource allocation scheme. We compare our method with existing large-scale training approaches and demonstrate its effectiveness by training a large language model using gaming GPUs in real-world internet conditions.

## 1 Introduction

As Large Language Models (LLMs) have gained popularity in recent years, many companies have invested in powerful AI GPUs, such as the A100, to train these models for months until convergence. The resources required to train such networks are immense, limiting this capability to only a few companies. For the vast majority of commercial GPUs, training an LLM is simply not feasible. Similarly, the bottleneck of computation and communication has hindered further development in these scenarios. Table 1 shows the computation cost of single-token inference and the approximate training time of well-known language models using mainstream GPUs Hobbhahn & Sevilla (2021); Casson (2023).

Existing works in distributed learning have thoroughly explored methods to run large transformer models on multiple GPUs. There are three main strategies to achieve this: model parallelism Dean et al. (2012); Shazeer et al. (2017), data parallelism Zinkevich et al. (2010); Goyal et al. (2017), and tensor parallelism Shoeybi et al. (2020); Narayanan et al. (2021a). Recently, hybrid pipeline approaches have been investigated Harlap et al. (2018); Fan et al. (2020). These works have significantly reduced the computation requirements for single GPUs and utilized GPU clusters to speed up training linearly. However, these studies primarily focus on large data centers with homogeneous GPU resources and high-bandwidth communication. Building and maintaining such infrastructure is notoriously costly, limiting access to only a few well-funded organizations. As a result, most researchers cannot afford to conduct the experiments required for a thorough evaluation of their ideas.

On the other hand, decentralization is a natural and promising direction. Research reports that the PC and Add-In Boards (AIB) GPU market shipped 9.5 million units in Q4 2023 alone wccftech. Furthermore, many of these GPUs are underutilized. If we could make use of these devices in a decentralized open-volunteering paradigm for foundation model training, this would be a revolutionary alternative to the expensive solutions offered by data centers. This vision has inspired many recent efforts in decentralized learning, encompassing both theoretical and algorithmic advancements Ren et al. (2021); Diskin et al. (2021); Atre et al. (2021).

Table 1: Computation cost (GMACs) of different models and time to train 100 epochs with an RTX 4070. GPU memory not considered.

| Model | Cost (GFLOPs) | Training Time (days) |
|---|---|---|
| ALBERT | 17.68 | 24.9 |
| BERT | 22.79 | 32.1 |
| LLAMA (70B) | 515.52 | 726.1 |
| GPT-3 (175B) | 1279.66 | 1902.2 |

However, training in such setups requires specialized algorithms that can adapt to the changing number of workers, utilize heterogeneous devices, and recover from hardware and network failures. While there are several practical algorithms designed for unreliable hardware Ryabinin et al. (2021), they can only train relatively small models that fit into the memory of the smallest device. This limitation reduces the practical impact of cost-efficient strategies, as today's large-scale experiments often involve models with billions of parameters.

Training deep learning models in distributed settings requires specialized algorithms that can adapt to varying numbers of workers, utilize heterogeneous devices, and handle failures. While existing methods provide solutions, they also have limitations: Moshpit SGD Ryabinin et al. (2022) efficiently manages unreliable devices, DeDLOC Ryabinin et al. (2021) supports heterogeneous training but is constrained by the memory capacity of smaller devices, and SWARM Ryabinin et al. (2023) accommodates unreliable devices and pipeline parallelism but lacks an efficient scheduler to optimize the pipeline. These constraints limit the practical impact of cost-efficient strategies in large-scale experiments.

In this work, we aim to find a solution to train large neural networks using **unreliable heterogeneous devices** with varying interconnects. We propose a framework that allows devices to **compute voluntarily**. We designate devices into a **reliable group** and an **unreliable group** and develop a **hybrid parameter server** Li et al. (2014) and pipeline parallelism approach. We provide a robust mechanism to correct and complete **leftover tasks**. Additionally, we analyze the latency and propose a heuristic approach to compute the **near-optimal network setup**.

In summary, we make the following contributions:

- We propose a method that allows devices to leave and join freely while providing fault tolerance for unstable connections. We leverage committed reliable devices and develop a learning scheme based on a parameter server and pipeline parallelism.
- We analyze the latency of our method, studying the time costs of data parallelism and pipeline parallelism under our scenario. We propose a novel optimization approach to provide a near-optimal allocation scheme for our network.
- We conduct extensive experiments to demonstrate the capability of our method in handling heterogeneous environments.

## 2 RELATED WORKS

### 2.1 PARALLELISM IN DISTRIBUTED LEARNING

Parallelization in machine learning is commonly categorized into data, operator, and pipeline parallelism Dean et al. (2012); Ben-Nun & Hoefler (2018). **Data parallelism** splits the training data across workers with model replication, allowing each worker to compute parameter updates independently, followed by synchronization to ensure consistent model parameters Zinkevich et al. (2010); Shallue et al. (2019). While effective for large datasets, data parallelism can be limited by memory constraints and synchronization overhead as models scale Jia et al. (2018). **Pipeline parallelism** divides the model into stages distributed across workers, processing microbatches in parallel Huang et al. (2019); Narayanan et al. (2021b). This method reduces memory requirements and pipeline bubbles but requires efficient allocation of each pipeline stage. **Operator parallelism** distributes computational operations, like matrix multiplications, across devices, which is particularly useful

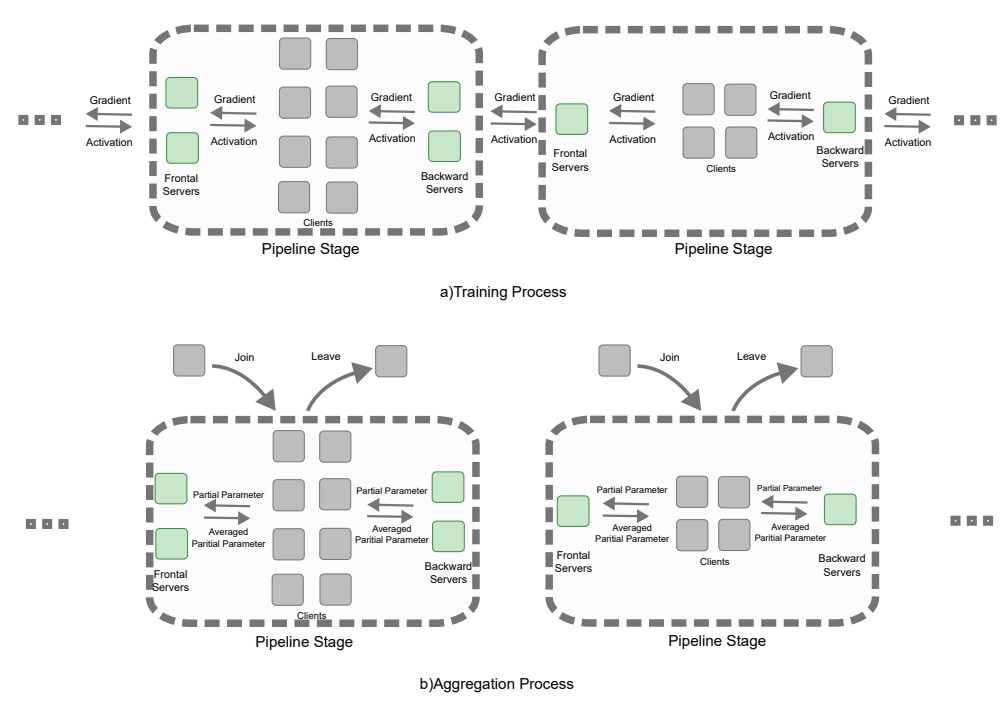

Figure 1: a) Training Process. Each pipeline stage perform pipeline parallel with parameter server in side each stage. b) Aggregation Process. Each pipeline perform sharded parameter aggregation using both FS and BS

for models with large layers Shoeybi et al. (2020); Wang et al. (2022). However, it can suffer from high communication costs and requires efficient partitioning. We do not consider it here because in our case, the system is highly sensitive to communication delays. Techniques like ZeRO-Offload Ren et al. (2021) and Megatron-LM Narayanan et al. (2021b) have demonstrated efficient scaling by addressing memory and communication challenges in training large models.

## 2.2 PARALLEL SCHEDULING

To implement parallel methods, a scheduling plan is needed based on hardware configuration and model structure. Scheduling can be either manual or automatic. **Manual** scheduling, like in Megatron, requires empirical adjustments at data, tensor, and pipeline levels. However, as device numbers grow, **automatic scheduling** becomes necessary. Studies such as Fan et al. (2020) have addressed this with dynamic and integer programming to optimize training time. Most focus on data center scenarios with homogeneous and reliable communication, while more recent work addresses heterogeneous communication Yuan et al. (2022) and stragglers Ryabinin et al. (2023); Diskin et al. (2021). However, these works only solve **part** of the problem. There remains a gap in addressing all the problems: distributed GPU resources across the unstable internet and growing model sizes.

## 3 METHODOLOGY

Motivated by the sub-optimalities observed in current architectures, this section introduces our proposed Methods. The framework consist of two parts: a optimization program and a runtime framework. The optimization program takes the device configuration and provide The runtime scheduler

is responsible for controlling each device to execute their assigned tasks. We first describe our setting in 3.1, then propose our workflow in 3.2, finally we solve the optimization in 3.3.

## 3.1 SETUP

### 3.1.1 HARDWARE SETTING

In an internet scenario, connections between devices are affected by geo-location and congestion, leading to heterogeneous throughput and temporary disconnections. Devices may have varying communication bandwidth and latency. Congestion can cause packet loss and random disconnections, which we assume are resolved within a reasonable time using TCP timeout and reconnection protocols.

We consider two groups of devices: reliable and unreliable. Reliable devices remain committed to the training process and attempt to reconnect after disconnections. Unreliable devices can join or leave the process freely. All devices have both CPU and GPU resources.

### 3.1.2 PARALLELISM SETTING

We work on transformer models, particularly those described in Vaswani et al. (2023). Transformer models are composed of multiple identical encoder or decoder layers, which can be divided into several stages for pipeline parallelism. Each stage consists of consecutive layers. In transformer models, the model dimension remains constant across all layers, meaning the activations and gradients communicated between each layer are of the same size. We also assume that the data fed into the network consists of millions of tokens, which can be split in any ratio as required for data parallelism. Given the two groups of devices and their communication topology, our method aims to allocate the model stages and minibatches on different devices to run distributed learning in both pipeline and data parallelism.

### 3.1.3 FORMALIZATION OF THE SCHEDULING PROBLEM

Here we formulate our problem formaly.

- For the reliable group, let $\mathbf{S} = \{s_1, \ldots, s_{N_s}\}$ be a set of $N_s$ devices; $\mathbf{C_s} = \{c_1^s, \ldots, c_{N_s}^s\}$ be the computation power of each device. Let $\mathbf{A_s} \in \mathbb{R}_+^{N_s \times N_s}$ and $\mathbf{B_s} \in \mathbb{R}_+^{N_s \times N_s}$ represent the communication matrices describing the delay and bandwidth between these devices, where the delay and bandwidth between device $s$ and $s'$ are $\alpha_{s,s'}^s$ and $\beta_{s,s'}^s$, respectively.

- For the unreliable group, let $\mathbf{U} = \{u_1, \ldots, u_{N_u}\}$ be a set of $N_u$ devices; $\mathbf{C_u} = \{c_1^u, \ldots, c_{N_u}^u\}$ be the computation power of each device. Let $\mathbf{A_u} \in \mathbb{R}_+^{N_u \times N_s}$ and $\mathbf{B_u} \in \mathbb{R}_+^{N_u \times N_s}$ represent the communication matrices between these devices and each reliable device, where the delay and bandwidth between unreliable device $i$ and reliable device $j$ are $\alpha_{i,j}^u$ and $\beta_{i,j}^u$, respectively.

- Given a neural network architecture $H$, the architecture can be divided into multiple stages $\{h_1, \ldots, h_K\} \in H$. In data parallelism, at the end of each epoch, parameters of $h_k$ are shared and averaged among all devices at the $k^{th}$ stage. Let $D$ be the number of tokens in each batch, and $d$ the number of mini-batches within each batch. Let $t$ be the size of activations and gradients for each token between stages. Thus, the size of total activations and gradients that need to be sent between adjacent devices is $\frac{tB}{d}$.

- Our goal is to allocate each $h_k$ to both reliable and unreliable devices such that the overall training time is minimized. The challenge is twofold: 1) How to schedule the pipeline and allocate each part of the model to each device to minimize the training time, and 2) How to effectively utilize the devices in the unreliable group in conjunction with the reliable group.

## 3.2 PROPOSED WORKFLOW

### 3.2.1 ASYNCHRONOUS PIPELINE SCHEDULING

Conventional pipeline parallelism may prioritize communication efficiency, but this alone is insufficient for our specific setup. In our scenario, training devices exhibit varying computational and network capabilities. Consequently, constructing a pipeline with such heterogeneous devices would encounter bottlenecks at the weakest participant, i.e., the node with the lowest training throughput. As a result, more powerful nodes in the pipeline would remain underutilized due to insufficient input or slow subsequent stages. Furthermore, any instance of node failure or premature departure from training would halt the entire training process.

To address the challenges inherent in conventional rigid pipeline stages, we propose leveraging a parameter server-based pipeline stage structure, as shown in Figure 1. Each stage in the pipeline incorporates a training network based on the parameter server model. Traditional parameter servers typically consist of one server and several clients, where each client sends training parameters and retrieves averaged parameters. In our framework, we introduce a partially-collocated parameter server architecture, operating across multiple parameter servers. All parameter servers belong to the reliable group $S$, while clients are drawn from the unreliable group $U$. In this configuration, parameter servers not only aggregate parameters but also relay activations and gradients among other parameter servers in adjacent pipeline stages. These parameter servers are categorized into frontal servers (FS) and backward servers (BS).

**Forward and Backward.** There are two processes in this scheme: the training process and the aggregation process. In the forward pass, shown in Figure 1, the frontal servers (FS) receive forward activations from their predecessors and distribute the activations to each client according to their relative computational power. Then, each client performs forward aggregation and sends the activations to the backward servers (BS). The BS passes the activations to the next stage while maintaining a list of which clients generated which activations. This list is implemented as a linked list, where each node contains the client ID (or IP address), starting index, and ending index of the activations. During the backward phase, the BS receives the gradients from the next stage and allocates them according to the linked list.

The transmission of activations and gradients between the frontal servers (FS) and the backward servers (BS) is evenly segmented based on the number of servers on the receiving side. For example, when two BS are sending activations to three FS, the first BS sends the first $\frac{2}{3}$ of activations to the first FS and the remaining $\frac{1}{3}$ to the second server. The second BS sends the first $\frac{1}{3}$ to the second FS and the remaining $\frac{2}{3}$ to the third FS. This segmentation ensures uniformity of indices for backward propagation while optimizing communication efficiency.

**Task Pool.** The motivation behind this is to leverage asynchronous training. If one client completes its task faster, it can request additional activations from the parameter server. Additionally, clients in this setting can freely join or leave. When a client leaves, other clients can take over the unfinished tasks. This is managed by maintaining a task pool in both the frontal servers (FS) and backward servers (BS). Whenever an FS detects that a client is unresponsive, it generates a forward task after a timeout and signals the other FS to add it to the task pool. Similarly, if the BS does not receive activations from a client, it posts the task to all FS. When any client completes its task, it requests the FS to take a new task from the task pool. Similarly, in the backward phase, when a client finishes working, it requests the BS for a task and asks the FS for the corresponding activation. Once it receives both the gradient and the activation, it recomputes the activation and completes the leftover task. This scheme ensures maximum throughput and allows training to continue even in the absence of unreliable clients.

**Aggregation Process.** In the aggregation process, shown in Figure 1b 1, we utilize Delayed Parameter Updates (DPU) Ren et al. (2021) to maximize efficiency. Each client sends its sharded parameters to both the frontal servers (FS) and backward servers (BS), processing them using the CPU, similar to the method in Jiang et al. (2020). Each client segments its parameters into the same number as the parameter servers. Then, the client sends its parameters to each parameter server accordingly, all in parallel. The parameter servers average the parameters on the CPU and send the averaged parameters back to the clients. DPU has demonstrated similar per-iteration convergence in theory Arjevani et al. (2020) and in experiments Diskin et al. (2021).

**Dynamic Client Participation.** In our scheme, client devices can freely join and leave the training process. When a client leaves the process, it can simply disconnect without any prior warning. The leftover tasks are posted to the task pool and executed by other clients. When a client wants to join, it first tests its connectivity to all parameter servers. It then averages the communication bandwidth and selects the pipeline stage with the highest bandwidth rate. Once selected, the client sends its computing power and IP address to all parameter servers in that pipeline stage. The parameter servers maintain and update a list that records all active clients and their computing power. At the end of each epoch, the parameter servers recompute the task segmentation based on the clients' computing power to balance the computation load.

## 3.3 LATENCY ANALYSIS

Scheduling in a heterogeneous setting poses a significant challenge due to the substantial expansion of the search space compared to homogeneous scenarios. We first model the training time of the data parallel method, and based on this result, we determine the total training time for the pipeline. The objective is to find the optimal partition of reliable and unreliable devices into multiple pipeline stages to minimize the overall training time. To address this problem, we design a multi-level approach based on graph theory.

### 3.3.1 MODELLING DATA PARALLELISM

Given a list of unreliable devices $\{u_1, u_2, \ldots, u_M\}$ and reliable devices $\{s_1, s_2, \ldots, s_N\}$, let submodel $h_k$ represent a pipeline stage where these devices perform data parallel processing together. The time cost for each iteration consists of three parts: receiving activations from the frontal servers (FS), performing forward (or backward) propagation, and sending the results to the backward servers (BS). For data parallel processing, there is also a cost associated with performing parameter aggregation. However, since we utilize the Delayed Parameter Updates (DPU) technique, this aggregation time is parallelized with the training process. Our experiments indicate that the aggregation time is less than twenty percent of the time required to train a single batch. Thus, the overall time cost can be modeled as follows.

$$DT_k = \max_{1 \le i \le M, 1 \le j < k < N} \left\{ \begin{array}{l} \alpha_{ij}^u + \frac{t_i}{\beta_{0,j}^u} + \alpha_{ik}^u + \frac{t_i}{\beta_{i,k}^u} + \frac{t_i |h_k|}{c_i^u} \\ \alpha_{jk}^s + \frac{t_i}{\beta_{jk}^s} + \frac{t_i |h_k|}{c_j^s} \end{array} \right. \tag{1}$$

Where $k$ is the current pipeline stage, and $t_i$ is the size of the activation assigned to the client.

Notice that the communication time of clients (the first four terms on the first line) can be minimized by assigning each client to the two parameter servers with the smallest average latency and highest communication bandwidth.

To minimize the computation time in the equation, two factors need to be considered. First, $h_k$ should be minimized, meaning that the number of pipeline stages, $K$, should be maximized. Secondly, the computational load $t_i$ must be balanced across all devices in proportion to their computation power. This is crucial because, compared to a perfectly balanced allocation, if one device runs faster while another runs slower, it increases the overall computation time.

### 3.3.2 MODELLING PIPELINE PARALLELISM

In our pipeline scheduling, we use the same 1F1B scheme proposed in DAPPLE Fan et al. (2020). In this scheme, the overall time consists of warming up, normal training, and ending phases. We can ignore the warming up and ending times due to their minimal contribution to the overall time. Now we approximate the overall pipeline time as follows:

$$PT = d \cdot \max_k \left( \alpha_{kf}^s + \frac{t_s}{\beta_{kf}^s} + \alpha_{kb}^s + \frac{t_s}{\beta_{kb}^s} + DT_k \right) \tag{2}$$

Where $k$ represents the pipeline stage, $\alpha_{kf}^s$ and $\alpha_{kb}^s$ are the delays between two adjacent pipeline stages in the forward and backward passes, and $\beta_{kf}^s$ and $\beta_{kb}^s$ are the corresponding bandwidths for forward and backward communication.

Note that the communication terms connecting predecessor and successor pipeline stages are additive to the communication term within the pipeline stage in the data parallel time, meaning that the three objectives are weighted equally for this problem.

### 3.3.3 SEARCHING PARTITION VIA GRAPH CLUSTERING

The objective is to find three partitions: the partition of the reliable group $\tau^*$, the partition of the unreliable group $\delta^*$, and the partition of model layers $\sigma^*$, in order to minimize the overall training time: $\min_{\tau,\delta,\sigma} PT$

Based on the results above, our approach first solves the partition of the reliable group, then allocates the unreliable devices to each partition according to communication characteristics. Finally, we determine the number of layers each partition should process based on its total computational power.

Previous works Fan et al. (2020); Zheng et al. (2022); Harlap et al. (2018) applied Dynamic Programming (DP) to solve the pipeline partition problem, given a list of devices and model layers. However, our scenario does not support such an approach because the graph structure lacks a defined axis and, therefore, does not exhibit the optimal substructure property.

To address the partition problem for reliable group formation, we create a communication graph $G$. Each node in $G$ corresponds to a device, and each edge between $s_i$ and $s_j$ is annotated with the averaged delay $\alpha_{ij}^s + \alpha_{ji}^s + \frac{t}{\beta_{ij}^s} + \frac{t}{\beta_{ji}^s}$.

Traditional graph clustering does not consider maximizing the number of stages as part of its target. Increasing the number of stages might also increase the communication time. Thus, we propose a modified hierarchical clustering of graphs Nielsen (2016). To maximize the number of clusters, $K$, we perform hierarchical clustering in a divisive (top-down) manner. At each level, we calculate the approximate pipeline time based on the current clustering result using Equation 2. Here, we approximate $DT_k$ as the average pipeline time:

$$DT_k \approx \frac{B|H|}{dp \sum C_s} \tag{3}$$

where $p$ is the current number of clusters. There are two reasons for this approximation: first, ideally, the pipeline time should be balanced across all stages, so we approximate the total time by dividing it by the number of clusters. Secondly, we discard the intra-cluster communication term since the goal of hierarchical clustering is also minimizing the distance within the cluster. We compare the pipeline time with the previous level, and if the pipeline time is higher at the current level, we select the previous level as the optimal clustering, $\tau^*$.

Once the partition of reliable groups is completed, we need to determine the order in which to assign each partition. The goal is to minimize communication between adjacent partitions. We average the edge labels between each partition and create a coarse graph $H$. The problem now becomes an Open Traveling Salesman Problem (OTSP) on the graph $H$. We then perform Simulated Annealing (SA) to solve this and find the optimal route across all partitions. Since the OTSP is solved on a partitioned graph, the scale of the problem is reduced.

Next, we assign each device to either the Frontal Server (FS) or the Backward Server (BS) in their respective partition groups. Devices are allocated based on which side (FS or BS) has the smallest average communication delay. Since FS and BS must have at least one device each, once all devices are assigned, we check the number of FS and BS. If FS has no device, we assign the slowest device from BS to FS, and vice versa.

Lastly, we allocate each unreliable device into partitions. Recall that we only need to minimize the delay according to Equation 1. We calculate the average delay $\frac{1}{|\tau_k|} \sum_{j \in \tau_k} \alpha_{ij}^u + \frac{t}{\beta_{ij}^u}$ for each partition $\tau_k$ and sort them. We then choose the partition with the smallest delay for each unreliable device, thus obtaining the partition of unreliable devices $\delta^*$.

Finally, based on the total computational power of each stage, we allocate model slices $h_k$ to each partition $\tau_k$. At this point, we have determined the partition for the model $\sigma^*$. To make the process easier to understand, we provide a toy example in Figure 6.

## 4 EXPERIMENTS

### 4.1 PERFORMANCE ANALYSIS

**Experiment Setup** In this experiment, we aim to compare our method with previous methods in both homogeneous and heterogeneous scenarios. We utilize a server cluster with 16 V100 GPUs to run the experiment. Servers at different geographic locations are used to simulate heterogeneous communication. We evaluate various configurations, using two transformer models: "Llama-33B" Touvron et al. (2023) and "ALBERT-xxlarge" Lan et al. (2019). The batch size is set to 512, and the minibatch size is set to 4 for both models. We measure the time taken per batch to compare overall training time, as all frameworks are equivalent in whole training process.

The purpose of this experiment is to compare training throughput under "ideal" conditions (homogeneous, reliable devices, and balanced layers). Deviations from these conditions make training with baseline systems infeasible. We compare our approach with PipeDream Harlap et al. (2018), Dapple Fan et al. (2020), and Swarm Ryabinin et al. (2023). Since Swarm does not provide an allocation algorithm, we use the same allocation as our optimized scheme for a fair comparison.

**Case 1: Single Datacenter**
We utilize 16 V100 GPUs with a 2GB/s inter-node connection, a common configuration in server environments. We do not control bandwidth or introduce disconnections in this secure datacenter setup.

**Case 2: Heterogeneous Communication**
GPUs in high-performance computing (HPC) systems are cost-effective but may be dispersed across different locations. Here, we deploy four HPCs, each with four V100s. However, some nodes have randomly initialized bandwidths (1-3 GB/s) and increased latency (10-100ms).

**Case 3: Heterogeneous Communication and Devices**
We combined two HPCs, each with four V100s, along with eight simulated RTX 2080 GPUs. Randomized communication patterns were introduced (throughputs between 50Mb/s and 300Mb/s, and latency between 100ms and 150ms) between the HPCs and standalone GPUs. To simulate the computational power difference between the RTX 2080 and V100 GPUs, we increased the training time of the RTX 2080 servers by 55%, proportional to the computation power ratio. This scenario evaluates scalability under heterogeneous computation and communication conditions.

**Case 4: Heterogeneous Scenario with Disconnection**
In this case, devices have varying setups and communication capabilities, with a 90 percent probability of sudden disconnection during training sessions, followed by reconnection in the next batch. This experiment assesses the fault tolerance of our algorithm.

Table 2 shows the results of our experiment. In Case 1 and Case 2, our method is slightly slower than PipeDream and DAPPLE because our method employs a parameter server approach at each stage, while PipeDream and DAPPLE use the Allreduce method for aggregating activations. In Case 3 and Case 4, our method outperforms the others due to the increased heterogeneity in communication and the presence of unreliable devices. The speedup can reach up to **1.5x** compared to DAPPLE. In Swarm, when a device leaves, the training data processed by that device is lost, requiring retraining. As a result, our method shows a **1.2x** speedup compared to Swarm.

### 4.2 UNRELIABLE DEVICE RATIO AND DISCONNECTION RATE

This experiment evaluates how varying the ratio of unreliable devices and their disconnection rates impact training throughput. We compare our method against Swarm and DeDLOC Diskin et al. (2021) using Llama. The cluster consists of 6 reliable devices (V100 GPUs) and 10 unreliable devices (RTX 2080 GPUs). We test five unreliable device ratios (20%, 30%, 40%, 50%, 60%) and five disconnection rates (20%, 30%, 40%, 50%, 60%). Throughput is measured in seconds per batch. Figure 2 summarizes the throughput times across all configurations. Our method consistently

Table 2: Comparison of Throughput Time for Different Systems

| | Throughput (min/batch) | | | | | | | |
| | "xxlarge" | | | | "Llama-33B" | | | |
| Model | Case 1 | Case 2 | Case 3 | Case 4 | Case 1 | Case 2 | Case 3 | Case 4 |
|---|---|---|---|---|---|---|---|---|
| Ours | 56.1 | 65.4 | **74.8** | **82.2** | 194.6 | 216.7 | **243.6** | **253.6** |
| PipeDream | **43.5** | 66.2 | 85.2 | 115.7 | 170.2 | 204.6 | 294.7 | 369.8 |
| Dapple | 44.3 | **55.9** | 79.1 | 114.6 | **168.9** | **185.2** | 268.5 | 356.4 |
| Swarm | 53.7 | 62.8 | 77.5 | 96.8 | 248.4 | 272.7 | 256.1 | 292.5 |

Figure 2: Time per batch for DeDLOC, Swarm, and our method across varying disconnection rates (10% to 90%) and unreliable device ratios (30%, 60%). Our method consistently shows better performance and resilience compared to baselines.

outperforms Swarm and DeDLOC with up to **1.3x** speedup over Swarm, especially under high disconnection rates and larger unreliable device ratios. This improvement is due to the fact that both baselines must reiterate the lost training data when the original device disconnects.

## 4.3 SCALABILITY AND HETEROGENEITY EXPERIMENT

In this experiment, we evaluate the scalability of our method under heterogeneous communication, devices, and unstable connections, scaling the number of V100 GPUs from 8 to 32. We compare our method with baseline methods, Swarm and Dapple, using Llama with a batch size of 512 tokens. The experiment includes four configurations (16, 24, 32, and 48 GPUs) under heterogeneous scenarios: devices experience different communication latencies (50–150 ms) and bandwidths (100–300 Mb/s) to simulate diverse network conditions. Unreliable devices have a 50–90% probability of disconnection during training, with reconnection in the next batch. We run this experiment 30 times, with our scheduler calculating an allocation scheme each time, and we report the averaged result. Figure 4 shows the throughput and scaling efficiency for each method across the different GPU configurations. Our method demonstrates higher and **near-linear** scalability, particularly under heterogeneous communication and unstable connections, compared to Swarm and Dapple.

## 4.4 REAL-WORLD TRAINING

In real-world settings, we apply our method across a diverse range of devices: an HPC with three RTX 4090s in the cloud, an HPC with three RTX 4090s within the campus network, and six RTX 3080s connected to the internet via a local ISP. We conduct pretraining using the Llama-33B model on WikiText-103 Merity et al. (2016). The hyperparameters are similar to those in the original paper Lan et al. (2019). Our goal is to test whether our setup is optimal.

First, we conducted a communication test, recording the end-to-end transmission bandwidth and latency of all device pairs, as shown in Figure 3. We used a customized VPN, modified from OpenVPN OpenVPN, to bypass Network Address Translation (NAT). We also used double precision and Zero-Offload to minimize VRAM requirements.

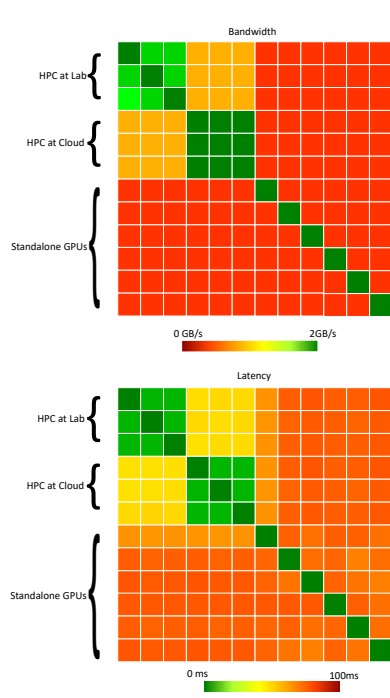

Figure 3: Communication and Latency matrix of real-world devices

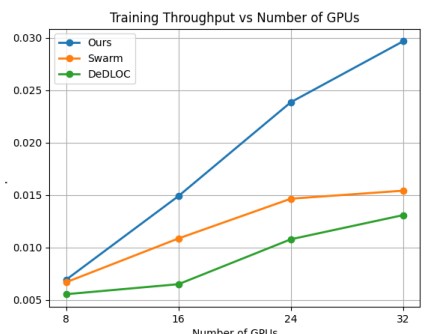

Figure 4: Training throughput in scalability experiment.

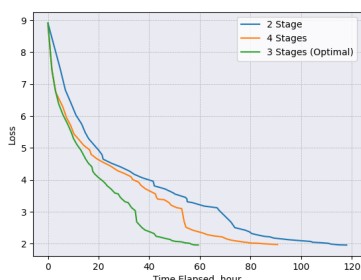

Figure 5: Training loss in real-world experiment

Among the six standalone GPUs, four are unreliable due to their intermittent usage. Using the communication matrices, we performed our optimization process and identified the optimal configuration, which consists of three stages: two stages for each HPC, and one stage for the reliable RTX 3080s, with one RTX 3080 assigned to each stage, respectively. For comparison, we set up configurations with four stages and two stages. The four-stage setup split the two HPCs into three stages. In the two-stage configuration, two RTX 3080s joined the campus HPC, and the other RTX 3080 joined the cloud HPC.

As shown in Figure 5, the three-stage configuration provided by our optimization algorithm **outperformed** both the two-stage and four-stage configurations. Our algorithm successfully leveraged the trade-off between communication and pipeline efficiency.

## 5 CONCLUSION

In this paper, we propose a Distributed collaborative learning framework. We combine parameter server and pipeline parallel, creating a distributed pipeline that support large scale transformer models training on unreliable devices. We carefully anaylized communication and computation cost and propose a heuristic graph clustering solution to provide optimal network allocation. We show that our method is effective compared to previous works and our scheduler provide an optimal configuration. As such, our work enables researchers without access to dedicated computing infrastructure to train large models.

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

## A APPENDIX

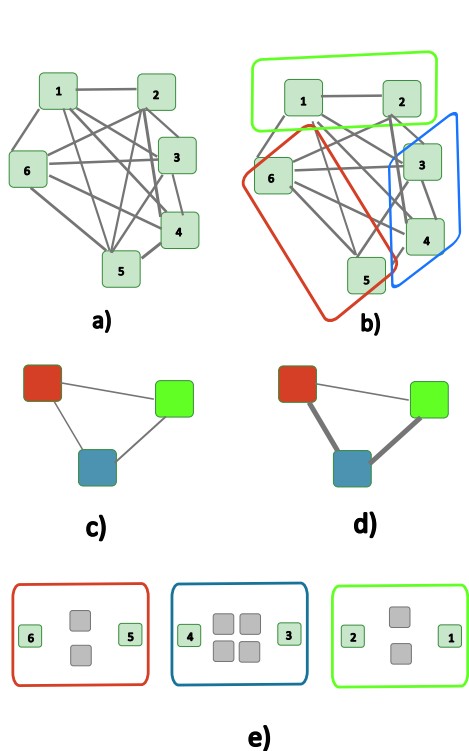

Figure 6: Toy example

