# OpenReview forum: "Decentralized Training of Transformer Models in Heterogeneous Network"
_ICLR.cc/2025/Conference — ICLR 2025 Conference Withdrawn Submission_

### Official Review · Reviewer_9oV6 · 2024-10-22

**Soundness:** 2
**Presentation:** 1
**Contribution:** 2
**Rating:** 3
**Confidence:** 4

**Summary:**

This paper proposes a distributed collaborative learning framework, which combines parameter server architecture and asynchronous pipeline parallelism to accelerate decentralized training on unreliable devices. Simulations and real experiments show that the proposed approach can outperform existing systems in decentralized, heterogeneous, and unreliable hardware settings.

**Strengths:**

1. The paper studies an interesting problem.
2. The experiments are conducted under different setups consisting of heterogeneous hardware and network conditions.

**Weaknesses:**

1. The experiments cannot show how the key design (i.e., asynchronous pipeline scheduling) affects the training efficiency. Using 16 V100 GPUs to train Llama-3.2-3B or 128 v100 GPUs to train Llama-33B would be more appropriate setups.
- ALBERT-xxlarge has only 223M parameters so pipeline parallelism is not necessary.
- Training Llama-33B with only pipeline parallelism requires more than 1000 GB of memory, and using 16 V100 GPUs means you offload more than half of the model to GPU memory or disk. In this configuration, the training bottleneck will shift to data movement between GPU and CPU/disk, making the impact of heterogeneous GPU and network conditions on training relatively small. It therefore cannot show the effectiveness of the proposed design under different conditions. In addition, if data parallelism is enabled as claimed in the paper, 2000 GB of memory is required even if the degree is only set to 2
2. More ablation studies are needed to investigate the performance improvements brought by scheduling for heterogeneous communication, scheduling for heterogeneous devices, and scheduling for unreliable devices respectively.
- Case 2 in Table 2 only involves heterogeneous communication. However, the scheduling strategy proposed in [1] can already solve this problem well.
- For Cases 3 and 4 in Table 2, it is better to show the performance of PipeDream, Dapple, and Swarm using existing scheduling strategies optimized for heterogeneous communication.
3. All of the figures are too small to read.
4. No source code is provided. Due to the huge amount of engineering effort to implement the proposed system, I would like to read the source code to understand the work better

[1] Decentralized Training of Foundation Models in Heterogeneous Environments

**Questions:**

How to integrate tensor parallelism and sequence parallelism into the framework?

---

### Official Review · Reviewer_FS2D · 2024-11-02

**Soundness:** 1
**Presentation:** 1
**Contribution:** 1
**Rating:** 1
**Confidence:** 4

**Summary:**

This paper presents an approach for decentralised training of deep learning models using affordable GPUs distributed across the globe, wherein the network latency and bandwidth between different devices can vary. The authors appear to be inspired by the work of Yuan et al. (2022), and trying to solve an optimisation problem to find the best mapping of pipeline stages to available devices.

**Strengths:**

Differently from the work of Yuan et al. (2022), this submission seems to be distinguishing between unreliable and reliable devices, where the reliable devices 1) act as parameter servers when exploiting data-parallelism and 2) relay activations and gradients between different pipeline stages.

**Weaknesses:**

- The paper appears to be solving an optimisation problem, yet the authors have not defined any decision (unknown) variables. Note that in  Yuan et al. (2022), the decision variables ($\sigma$) were introduced to define the mapping between the tasks and devices.

- What $t_i$ and $t_s$ represent is not clear. It is stated that "$t_i$ is the size of the activation assigned to the client." Which client? Is $i$ the index of the client, index of a task, or the index of a device. How are $t_i$ and $t_s$ computed? Are they known or unknown variables?

- The authors use a clustering algorithm followed by some post processing. However, there is no formal description or pseudocode of the algorithm.

**Questions:**

- What are the unknown (decision) variables and known variables used in the optimisation?
- Can you report some numbers on the runtime of your optimisation algorithm?
- How does your optimisation algorithm deal with failures in unreliable devices? Do you have to run optimisation from scratch to remap the failing tasks? Can this be done online?

---

### Official Review · Reviewer_RcT7 · 2024-11-04

**Soundness:** 2
**Presentation:** 1
**Contribution:** 1
**Rating:** 3
**Confidence:** 4

**Summary:**

The paper addresses the challenge of training large transformer models in decentralized and heterogeneous environments. It improves on the existing framework by considering both unreliable and reliable devices for model training (eg. parameter servers, pipeline parallelism), and task pool strategies. The authors introduce latency and time-cost analysis to optimize resource allocation, achieving performance gains even with unstable communication conditions. The work is evaluated through experiments under various heterogeneous and unreliable setups.

**Strengths:**

* The paper introduces a hybrid parallelism method with parameter servers to address the inefficiencies in existing decentralized frameworks, demonstrating the possibility of using geo-distributed heterogeneous devices to train foundation models.
* The experiments cover multiple scenarios, including varying device reliability and communication bandwidths, providing strong empirical support for the proposed framework.
* The research is highly relevant given the growing interest in decentralized and cost-effective training methods for large models, offering a potential alternative to expensive data center-based training.

**Weaknesses:**

* The proposed framework bears significant conceptual and methodological overlap with prior research, particularly in scheduling and optimization techniques. A clearer delineation of novel contributions is needed.
* While the method performs well in tested scenarios, the complexity of the proposed solution and its scalability to larger, more variable setups could be discussed in more detail.
* The writing of this paper could be improved by replacing the mathematical formulations with textural descriptions or figure illustrations.

**Questions:**

Please clarify the novelty of this submission apart from the existing work "Decentralized Training of Foundation Models in Heterogeneous Environments". Specifically,
* Both papers discuss using multiple forms of parallelism, specifically data and pipeline parallelism, to manage the distribution of tasks among heterogeneous devices. What are the differences in the types of parallelism considered here?
* Both papers describe using graph-theoretic approaches, such as graph clustering and hierarchical partitioning, to optimize task allocation and reduce communication costs. What are the differences in the formulation of the scheduling problem and the methodology used to address it?
* Could you share if this work builds upon or significantly differs from other frameworks such as those mentioned in the related work (e.g., DeDLOC, PipeDream, DAPPLE, DT-FM)?

---

### Official Review · Reviewer_81z8 · 2024-11-08

**Soundness:** 2
**Presentation:** 2
**Contribution:** 1
**Rating:** 1
**Confidence:** 5

**Summary:**

The paper proposes a framework for training large transformer-based LLMs on a heterogeneous network of widely distributed GPUs, which are typically more affordable but underutilized. It aims to optimize resource allocation through a mix of parameter server architectures, pipeline parallelism, and a dynamic task pool to manage unstable connections and device variability.

**Strengths:**

Leveraging underutilized, globally-distributed GPUs for training large-scale models looks promising and useful, especially for many individuals and organizations who cannot afford specialized clusters.

**Weaknesses:**

* **Unrealistic Assumption**: This proposed approach relies on the assumption that the devices with more stable network connections are reliable and can be served as the parameter server. However, this assumption may not be true in practice since the GPU failure may still happen and lose all model parameters even have a reliable network connection.
* **Limited Technical Novelty**: The proposed asynchronous pipeline scheduling method basically just describes the asynchronous parameter server approach in prior work, which has been widely studied in related work. It's unclear to me that which technical part is the key contribution of this paper and how it is different from existing techniques.
* Please address all writing issues (e.g., wrong paper title, too small font size in figures, and citation format) to make the paper more readable.

**Questions:**

What are the key differences compared with prior work like SWRAM?

---

### Comment · Area_Chair_EURF · 2024-11-21
**No author response yet**

Dear Submission9467 Authors,

ICLR encourages authors and reviewers to engage in asynchronous discussion up to the 26th Nov deadline. It would be good if you can post your responses to the reviews soon.

---

### Note · Authors · 2024-11-24

I have read and agree with the venue's withdrawal policy on behalf of myself and my co-authors.